# Bta-miR-6531 Regulates Calcium Influx in Bovine Leydig Cells and Is Associated with Sperm Motility

**DOI:** 10.3390/genes13101788

**Published:** 2022-10-03

**Authors:** Qiang Ding, Xiuhu Ding, Shuwen Xia, Fang Zhao, Kunlin Chen, Yong Qian, Shaoxian Cao, Zhiping Lin, Yundong Gao, Huili Wang, Jifeng Zhong

**Affiliations:** 1Institute of Animal Science, Jiangsu Academy of Agricultural Sciences, Nanjing 210014, China; 2Key Laboratory of Crop and Animal Integrated Farming, Ministry of Agriculture and Rural Affairs, Jiangsu Academy of Agricultural Sciences, Nanjing 210014, China; 3College of Animal Science & Technology, Nanjing Agricultural University, Nanjing 210095, China; 4Jiangsu Youyuan Dairy Research Institute Co., Ltd., Nanjing 211106, China; 5Shandong OX Livestock Breeding Co., Ltd., Jinan 250131, China

**Keywords:** sperm motility, bta-miR-6531, *ATP2A2*, calcium signaling pathway, SNP

## Abstract

MicroRNAs (miRNAs) play key roles in sperm as the regulatory factors involved in fertility and subsequent early embryonic development. Bta-miR-6531 is specifically a highly enriched miRNA in low-motility sperms in previous study. To investigate the mechanism of bta-miR-6531, 508 shared target genes of bta-miR-6531 were predicted using two miRNA target databases (TargetScan7 and miRWalk). According to the Kyoto Encyclopedia of Genes and Genomes (KEGG), the calcium and cAMP signaling pathways were the most enriched of the target genes. A dual-luciferase assay indicated that bta-miR-6531 targeted *ATP2A2* mRNA by binding to the coding sequence region. In bovine Leydig cells, bta-miR-6531 overexpression affected the intracellular calcium concentration by restraining *ATP2A2* expression. Moreover, we observed high calcium concentrations and high *ATP2A2* protein levels in high-motility sperm compared with those in low-motility sperms. Furthermore, high-linkage single-nucleotide polymorphisms (SNPs) of the pre-bta-miR-6531 sequence were identified that related to sperm traits. Genotype TCTC of bta-miR-6531 showed high sperm motility and density and low deformity rate in Holstein bulls. However, the mutation in pre-miR-6531 did not significantly affect mature bta-miR-6531 expression in the sperm or cell models. Our results demonstrate that bta-miR-6531 might involve in sperm motility regulation by targeting *ATP2A2* of the calcium signaling pathway in bovine spermatozoa.

## 1. Introduction

MicroRNAs (miRNAs), a class of short single-stranded non-coding RNAs of with a length of approximately 18–24 nt, are known to negatively regulate mRNA expression or repress protein translation by binding to mRNA transcripts [1]. Several studies have reported that sperm miRNAs play essential roles in the fertilization of zygotes and in early embryonic development [2,3,4]. Deep sequencing has identified thousands of miRNAs that are differentially expressed between low- and high-motility sperm in cattle [2,3,5].

Canonical biogenesis pathways of miRNAs have been defined. In the nucleus, miRNAs are initially transcribed from long primary transcripts (termed pri-miRNAs) and further processed into precursor miRNAs (pre-miRNAs) by RNase Drosha [6]. Then, the pre-miRNAs are exported to the cytoplasm by Exportin 5 and further cleaved into mature miRNAs by DICER [1]. Mature miRNAs recognize their target mRNAs mainly by the base-pairing interactions between nucleotides 2–8 (seed region) from the 5′ end and complementary nucleotides in the 3′ untranslated region of the target gene mRNA. Thus, it is estimated that a miRNA may target hundreds of genes in the mammalian transcriptome and many miRNAs regulated most genes.

Sperm quality is the most important index in the selection of breeding bulls. Sperm quality traits are evaluated using several parameters, including sperm motility, density, viability, and morphology. Sperm motility is a major trait with high heritability (close to 0.60) in bull breeding programs [7]. With the development of molecular marker techniques and marker-assisted selection in animal breeding, single-nucleotide polymorphisms (SNPs) in candidate genes could be used as markers to predict sperm quality traits in bulls. Over the past two decades, the number of SNPs in quantitative trait loci (QTL) has been related to sperm quality in cattle [7,8]. However, the genetic potential of the reproductive characteristic of Holstein bulls has not been fully explored.

miRNA-related SNPs have been demonstrated to have complex effects by affecting the miRNA function, including miRNA maturation, functional strand selection, and target gene selection [9]. Mutations in the miRNA precursor sequences potentially cause mammalian diseases and influence biological traits [10,11]. Based on our previous findings, we used a customized miRNAQTLsnp software to identify several SNPs loci in bovine miRNA such as pre-bta-miR-6531 [12]. These SNPs were also found to be co-located at the quantitative trait loci (QTLs) that may be associated with semen quality.

Therefore, in the present study, we aimed to elucidate the molecular mechanism of miR-6531 regulation of sperm motility in cattle, and further confirm whether the candidate SNPs in the bta-miR-6531 precursor sequence are related to sperm quality.

## 2. Materials and Methods

### 2.1. Ethics Statement

Animals were not used in this study. All Bovine semen was provided by the Shandong OX Livestock Breeding Co., Ltd. (Jinan, China).

### 2.2. Bta-miR-6531 Target Gene Prediction and Kyoto Encyclopedia of Genes and Genomes (KEGG) Analysis

Based on the bta-miR-6531 sequence provided by the miRbase [13], two miRNA databases containing the cow database, TargetScan (http://www.targetscan.org/, accessed on 29 September 2021) [14] and miRWalk (http://mirwalk.umm.uni-heidelberg.de/, accessed on 29 September 2021) [15], were used to predict bta-miR-6531 target genes. The shared genes of the two databases were further used to perform KEGG pathways enrichment analyses using KOBAS software (KEGG Orthology-Based Annotation System [http://kobas.cbi.pku.edu.cn/, accessed on 29 September 2021]). Significantly enriched pathways were identified using hypergeometric distribution and Fisher’s exact test at 5% level of significance (*p* ≤ 0.05).

### 2.3. Reporter Plasmid Construction and Dual-Luciferase Reporter Assay

The target gene *ATP2A2* CDS region sequences harboring the binding seed region of bta-miR-6531 were amplified using primers containing two restriction sites, which are listed in Appendix A. PCR products were isolated and inserted into the pmirGLO vector, following the T4 ligase according to the manufacturer’s instructions. Mutations in the seed regions were constructed via site-directed mutagenesis using the Mut Express II Fast Mutagenesis Kit V2 (C214-01, Nanjing Vazyme Biotech Co., Nanjing, China).

HeLa cells were seeded in a 12-well plate and cultured in DMEM/F12 (SH30023.01B, HyClone, Logan, UT, USA) medium supplemented with 100 U/mL penicillin, 100 µg/mL streptomycin (ThermoFisher), and 10% fetal bovine serum (FBS [HyClone, Marlborough, MA, USA]) at 37 °C and 5% CO_2_. After 24 h, when adherent cells reached 70% confluence, the pmirGLO-targets-3′UTR or mutagenesis vectors (400 ng), synchronically with miR-6531 mimics or negative control, were co-transfected into cells according to the Lipofectamine3000 (Invitrogen, Waltham, MA, USA) protocol. After 48 h, the dual-luciferase assay system (Promega, corporation, Madison, WI, USA) was used according to the manufacturer’s instructions. Cells were lysed and fluorescence intensity was detected using a multifunctional microplate reader (GLOMaX-Muti Plus [Promega Corporation, Madison, WI, USA]). Firefly luciferase activity was normalized to Renilla luciferase activity in each transfected well. All experiments were repeated three times.

### 2.4. Separation, Culture, and Identification of Bovine Leydig Cells

Leydig cells from bovine testes were separated using density gradient centrifugation as previously reported [16]. Briefly, testis tissues were digested with collagenase IV at 37 °C for 30 min. The digestion mixture was centrifuged at 800× *g* for 5 min, and the supernatant was discarded to remove the cellular debris. Furthermore, the cell pellet was layered on different Percoll density gradients (70%, 55%, and 35%) and centrifuged at 2000× *g* for 1 h at 4 °C. The cells in 55% Percoll were collected and resuspended in Dulbelcco`s Phosphate Buffer Saline (HyClone, Logan, UT, USA) to remove the remaining Percoll. Cells were cultured in DMEM/F12 medium supplemented with L-glutamine, 10% FBS and nonessential amino acids at 34 °C and 5% CO_2_. The purity of the Leydig cells was determined using the immunohistochemical of the 3 β-hydroxysteroid dehydrogenase (3β-HSD).

### 2.5. Measurement of Calcium in Transfected miR-6531 Bovine Leydig Cells

The fluorescent calcium chelating dye Fluo-4 AM was used as an indicator of the relative levels of intracellular calcium in Leydig cells transfected with NC mimics or bta-miR-6531 mimics in Confocal Dishes (NEST). Cells were incubated with Fluo-4 AM (2 μM) diluted in Hank’s balanced salt solution (HBSS, calcium, and magnesium free, HyClone) for 30 min at room temperature. After Fluo-4 AM was loaded, the cells were washed with HBSS buffer to remove residual Fluo-4 AM. The fluorescence of calcium was captured by a fluorescence microscopy at 100 ms exposure. The intensity of calcium signaling was measured by using Image Pro-Plus 6.0 and remove the background. For flow cytometric analysis of calcium, cells were digested by 0.25% trypsin and resuspend before load Flou-4 AM. Cell calcium flux were stimulated by ionomycin (2 μM).

### 2.6. Total RNA Isolation, cDNA Synthesis, and qPCR

The transcriptomes of sperms and cells were isolated by using TRIzol reagent (15596018, Ambion Inc., Austin, TX, USA) according to the protocol of the PureLink RNA Mini Kit (12183018A, Ambion Inc.). RNA extraction from sperms were performed using guanidinium thiocyanate supplemented with Tris(2-carboxyethyl)-phosphine (646547-10 × 1 mL, Sigma-Aldrich) [17]. For mRNA expression analysis, first strand cDNA was synthesized using HiScript^®^ III 1st Strand cDNA Synthesis Kit (R111-01, Nanjing Vazyme Biotech Co.) with random hexa-primers. qPCR reactions were performed using ChamQ SYBR Color qPCR Master Mix (Q411-02, Nanjing Vazyme Biotech Co.) on QuantStudio 5 (Thermo Fisher Scientific, Waltham, MA, USA). Glyceraldehyde 3-phosphate dehydrogenase (*GAPDH*) was used as a reference gene to normalize reactions in mRNA analysis. The relative expression of each sample was calculated using the 2^−ΔΔCt^ method [18]. The primers used for qPCR are listed in Appendix A.

### 2.7. Protein Extraction and Western Blotting

For protein extraction, cells were incubated in radioimmunoprecipitation assay (RIPA) lysis buffer (P0013B, Beyotime Biotechnology, Shanghai, China) supplemented with a protease and phosphatase inhibitor cocktail (P1051, 50×, Beyotime Biotechnology) and placed on ice for 30 min. For sperm protein analysis, the sperm samples were selected based on different motility (Appendix A). 1% Sodium dodecyl sulfate was added to the RIPA lysis buffer. Sperm samples were placed on ice under ultrasonic conditions for 10 cycles of 5 s pulse with a 30 s interval. The protein concentrations of all samples were measured by using the BCA assay (P0010S, Beyotime Biotechnology). Protein samples were mixed with Sodium dodecyl-sulfate polyacrylamide gel electrophoresis (SDS-PAGE) loading buffer (P0015L, Beyotime Biotechnology) and boiled for 10 min.

Additionally, 20 μg total proteins was loaded onto a 12% SDS-PAGE gel for separation based on the protein size. The proteins were then transferred onto a polyvinylidene difluoride (PVDF) membrane (Roche, Mannhein Germany) at 260 mA for 1 h. The membranes for immunoblotting were blocked with 5% non-fat milk powder at room temperature for 2 h and incubated with primary antibodies (*ATP2A2*, 1:500; *ATP1A1*, 1:500; β-actin, 1:2000) at 4 °C overnight. Secondary antibodies (anti-mouse 1:2000 or anti-rabbit 1:2000) were used depending on the primary antibody. β-actin was used as a reference protein to measure the relative expression of target proteins.

### 2.8. Sperm Samples

The Holstein bulls in the present study ranged between 2.5 and 11 years old and were obtained from the Shandong bull station (Jinan, China). Fresh semen samples were collected using a bovine artificial vagina, quickly transferred to the laboratory, and incubated at 37 °C after collection from each bull. Subsequently, the semen volume, sperm concentration and sperm motility were determined. Ejaculate volume was calculated using a collecting vial, the density of fresh sperms was determined using a hemocytometer. Semen was packaged in 0.25 mL straws after diluting with Bioxcell (IMV Biotechnology, Basse-Normandie, France) and cryopreserved. After storage in liquid nitrogen 7 days, 2 straws were randomly selected from each ejaculate, thawed at 38 °C for 20 s, and immediately evaluated for sperm motility and deformity rate. For motility of fresh or post-thawed sperm, a drop of semen was placed on a pre-warmed (37 °C) glass slide and covered with a glass slip on a thermo plate. The sperm motility was evaluated by using a phase-contrast microscope at 400× *g* magnification on a TV monitor with a sperm analysis system (AndroVision Minitube, Germany). For sperm motility calculate, all processive movement or quickly movement sperms were considered as motile sperms in this study.

The percentage of strawed sperm deformities was evaluated using Giemsa staining solution (Beijing Solarbio Science & Technology Co., Ltd., Beijing, China) at 400× *g* and 1000× *g* magnification [19], with more than 200 stained sperms in consideration. Some morphology defects, coiled or hairpin or terminal droplet tail, detached head and no or small acrosome were considered as abnormal sperm in the present paper.

### 2.9. SNP Identification and Genotype Analysis

Our previous study identified the sequence of pri-miR-6531 containing SNPs located in QTLs associated with sperm motility in Holstein bulls. Therefore, we amplified the pri-miR-6531 sequence and performed Sanger sequencing. Fifty-six bulls with semen parameter records were used in the association analysis of the SNPs of pri-miR-6531 and semen quality traits. Semen quality traits were collected and provided by Shandong OX Animal Breeding Co., Ltd., Shandong, China. The genomic DNA of each sperm sample was isolated from the frozen semen according to a previous report [8]. The pair of specific primers for the pri-miR-6531 sequence is listed in Appendix A.

### 2.10. Pre-bta-miR-6531 Structure Prediction and bta-miR-6531 Expression Analysis

The structures of the miRNA precursors were determined by using mRNA structure software (RNAstructure, Version 6.4, Developed by Jessica S. Reuter & Maintained by Richard M. Watson, New York, USA). For analysis of the bta-miR-6531 expression level, cDNA and quantitative PCR (qPCR) were performed using miRNA 1st Strand cDNA Synthesis Kit (by stem-loop [MR101-01, Nanjing Vazyme Biotech Co., Nanjing, China]) and miRNA Universal SYBR qPCR Master Mix (MQ101, Nanjing Vazyme Biotech Co., Nanjing, China) according to the manufacturer’s instructions. U6 was used as the reference miRNA for normalization, and the fold change relative to the control samples was determined by the 2^−ΔΔCt^ method. All primer pair sequences are listed in Appendix A.

### 2.11. Statistics Analysis

All data are shown as the means ± SEM. Experiments for each group were repeated at least three times. A two-tailed Student’s *t*-test was used for group comparisons. Three or more groups were compared using one-way analysis of variance and Tukey’s test. All statistical graphics were drawn in GraphPad Prism 9 software. Differences of *p* < 0.05 and differences of *p* < 0.01 were considered statistically significant and highly statistically significant, respectively.

## 3. Results

### 3.1. miR-6531 Targets the Calcium Signaling Pathway and the cAMP Signaling Pathway

To further explore the expression of bta-miR-6531 in bovine sperm, sperm with high or low motility were separated. qPCR analysis showed that bta-miR-6531 was lost in sperm with high motility (Figure 1A). To understand the function of bta-miR-6531, we evaluated its cellular targets. Thus, 2349 and 2739 target genes were predicted using miRWalk and TargetScan 7, respectively. Among all the target genes, 508 were shared in the two miRNA databases (Figure 1C). The KEGG pathway of shared targets genes showed that the most enriched pathways were axon guidance, cAMP signaling pathway, calcium signaling pathway, and thyroid hormone signaling pathway, all of which are actively involved in sperm (Figure 1B).

### 3.2. miR-6531 Regulated Calcium Flex into Cells

Pathway enrichment analysis showed that bta-miR-6531 directly targeted the calcium signaling pathway and the cAMP signaling pathway. We further performed intracellular calcium concentration [Ca^2+^]_i_ assays and mitochondrial membrane potential (MMP) in Leydig cells. The fluorescence intensity of [Ca^2+^]_i_ in bta-miR-6531-overexpressing cells showed a low concentration of [Ca^2+^]_i_ compared with NC groups (Figure 2A). The results of flow cytometry of the [Ca^2+^]_i_ signaling showed a low level after overexpressing bta-miR-6531, but present same level after stimulated by ionomycin (Figure 2B). Additionally, overexpression of bta-miR-6531 enhanced the MMP in Leydig cells (Appendix A), indicating that bta-miR-6531 plays a biological role in regulating calcium in bovine Leydig cells.

### 3.3. Bta-miR-6531 Binds and Downregulates ATP2A2 Expression in Bovine Leydig Cells

Bioinformatic analysis showed that bta-miR-6531 targets the *ATP2A2* gene and identifies the existence of a putative bta-miR-6531 binding site within the *ATP2A2* CDS region. To confirm whether bta-miR-6531 directly binds to the *ATP2A2* CDS region, luciferase reporter vectors expressing the bta-miR-6531 binding site in the *ATP2A2* CDS region or mutated binding site were transfected into HeLa cells and the relative luciferase activity was assessed (Figure 3A). The relative luciferase activity of wild-type reporters was significantly reduced in cells co-transfected with bta-miR-6531 mimics compared with that of NC mimics, whereas the mutant reporter vector was unaffected (Figure 3B).

To confirm the target genes affected by bta-miR-6531, mRNA and protein expression were detected in Leydig cells after overexpression or interference of miR-6531 (Figure 3C). In the Leydig cell model, overexpression of bta-miR-6531 did not markedly affect the mRNA expression of target genes; however, we observed that the *ATP2A2* protein was significantly upregulated after the overexpression of miR-6531 compared with that of NC mimics. Contrastingly, the inhibition of miR-6531 expression resulted in a very low *ATP2A2* protein expression level compared with that of the NC inhibitor (Figure 3C).

### 3.4. ATP2A2 Different Expression Effeceted Calcium Loading into Bovine Leydig Cells

To investigate the calcium flux in cells whether affect by *ATP2A2*, we designed small interfere RNA (siRNA) of target CDS region of bta-miR-6531. The results showed that the Ca intensities were lower in cells after knockdown the *ATP2A2* expression than negative control (Figure 4A). To further explore the relationship between miR-6531 and the expression of its target genes in cattle sperm, we extracted proteins from sperms with high or low motility. The Western blotting results showed that *ATP2A2* protein was significantly highly accumulated in sperm with high-motility than in low motility (Figure 4B). The above results demonstrated that *ATP2A2* may involve in regulate sperm motility by regulate calcium influx.

### 3.5. Sequencing Analysis and SNPs Detection

PCR products were sequenced using specific primers to align the sequences of the bta-miR-6531 precursor. SNP data were created from sperm DNA samples from 56 bulls by Sanger sequencing. Sequence alignment revealed four SNPs (A-T, T-C, C-T, and T-C) located on the sequence of the bta-miR-6531 precursor, specifically, downstream of the mature sequence of bta-miR-6531 (Figure 5A). The structure of pre-miR-6531 was predicted by RNA fold [20]. The results showed that mutations in the precursor did not considerably affect the pre-miR-6531 structure (Figure 5B). Furthermore, the locations of the four SNPs at the genomic level were close, and linkage disequilibrium analysis showed that they were strongly linked (Figure 5C).

To identify whether mutations within the precursor of bta-miR-6531 affect mature miRNA expression, we constructed two genotypes of miRNA precursor overexpression vectors and transfected them into bovine Leydig cells. qPCR results showed that mutation of the miR-6531 precursor did not markedly influence mature bta-miR-6531 expression (Figure 5D), indicating that the mutation of pre-bta-miR-6531 is not a functional change but could be used as a marker for sperm quality.

### 3.6. Linkage between SNPs of the Pre-bta-miR-6531 is Associated with Sperm Quality

We analyzed the effects of this genetic variation on sperm quality traits by using SPSS18.0 software (IBM, Armonk, NY, USA). Multiple comparisons were performed using Tukey’s test. It was expected that bulls with the TCTC genotype would show higher fresh sperm motility and sperm density, and a lower deformity rate than those with the ATCT genotype (*p* < 0.05). There were no significant differences in ejaculate volume, and post-thaw cryopreserved sperm motility between the genotypes (*p* > 0.05) (Table 1), indicating that bulls with the TCTC genotype are more likely to show high sperm quality.

## 4. Discussion

Numbers of miRNAs were identified different expressing between high- and low-motility sperms. Bta-miR-6531 is special low expressing in low-motility sperms, the potential molecular function of miR-6531 in sperm has not been elucidated. It was observed that predicted genes of bta-miR-6531 mainly target the calcium and the cAMP signaling pathway. The influence of calcium and cAMP on sperm motility and the relationship between semen parameters and cAMP remains controversial [21]. Moreover, in the bovine Leydig cells model, overexpression of bta-miR-6531 resulted in lower [Ca^2+^]_i_ and higher MMP than that of the control group. miR-6531 is a retentive miRNA in sperm, that may regulate spermiogenesis in bovine testis. The generation of cAMP can stimulate influx through specific calcium channels, contributing to sperm hyperactivation [22]. Sperm hyperactivation is required to penetrate of the zona pellucida, which is critical for fertilization. As in most cell types, calcium is a key regulator of biofunctions.

*ATP2A2*/*SERCA2* is present in mammalian sperm and detected in the acrosome and midpiece areas [23]. During spermatogenesis, *ATP2A2* protein expression begins at the primary spermatocyte stage and maintains high signaling levels in the round and elongated spermatids [23]. Typically, *ATP2A2*/*SERCA2* regulate intracellular calcium into endoplasmic reticulum in cells [24]. The level of *ATP2A2*/*SERCA2* of sperm may related the calcium flex. Intracellular calcium is a key regulator of the sperm physiology, including sperm maturation, sperm motility, capacitation and acrosome reaction [25,26]. In present study, we have identified *ATP2A2* as a functional target gene of miR-6531 by targeting its CDS region in bovine Leydig cell. *ATP2A2* is present in the head and midpiece of the sperm [23], but the functional role of *ATP2A2* in sperm are not exactly clear. In this study, we verified a possible relationship between miR-6531 and *ATP2A2* in a somatic cell model. We also constructed a linkage between miR-6531, *ATP2A2* and calcium in bovine sperm. But this relationship whether occurs accurately in the sperm is still unclear, and the sperm-borne miRNAs might show a little contribution to sperm motility, we just provided a probable event between miR-6531 and *ATP2A2*. Many extensive researches of expression changes of sperm-borne miRNAs related to sperm quality [27]. Studies on sperm-borne miRNAs have mostly focused on the subsequent fertilization and embryonic development [3,28,29]; thus, the miRNA biological functions in sperm need to be explored in more studies.

Genomic SNPs associated with sperm quality have been reported in cattle [8,30] Several functional SNPs in the bta-miR-6531 precursor that are related to sperm quality and its possible molecular mechanism in sperm have been identified. In general, two types of functional SNPs occur in miRNA. Firstly, mutated SNP in target sites might lose the seed region, which is required for binding to target genes [31]. Secondly, SNP may influence the structure of the pre-miRNA, leading to changes in the biogenesis of mature miRNA, effecting the miRNA expression level [9]. The identified variations in pre-bta-miR-6531 in the present study is not located within the target site and did not considerably affect the expression of mature bta-miR-6531 in the sperm with different genotypes. However, bulls with the TCTC genotype showed high fresh sperm motility and density, and a lower deformity rate than that of ATCT genotype. This might indicate that these SNPs are likely associated with better sperm quality.

In conclusion, this study revealed miR-6531 regulate calcium signaling by targeting *ATP2A2* in bovine Leydig cells and identified a series of SNPs in the precursor of bta-miR-6531 in Holstein bulls, that can be used as molecular biomarkers associated with sperm quality. Also provides an opportunity for a more detailed investigation of the contribution of miRNA-associated SNPs to sperm quality.

## Figures and Tables

**Figure 1 genes-13-01788-f001:**
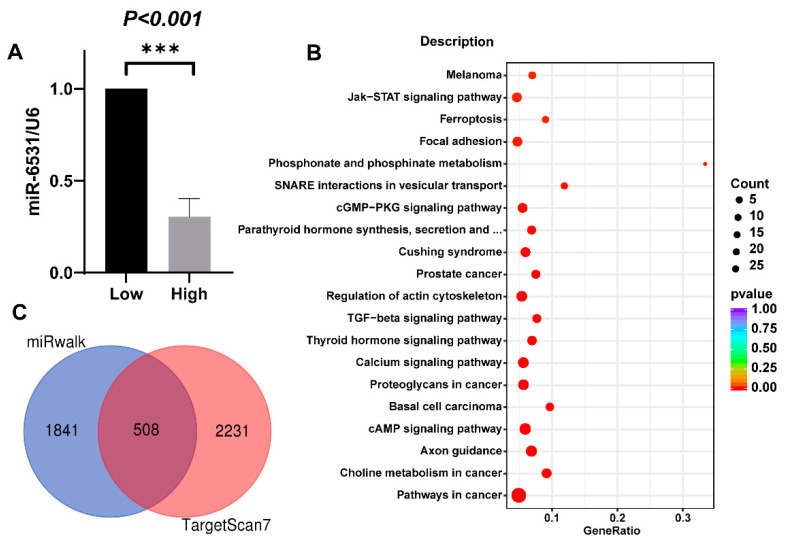
The predicted targets of bta-miR-6531 are mainly involved in the calcium signaling pathway and the cAMP signaling pathway. (**A**) The expression of miR-6531 in different motility sperms; (**B**) Targets of bta-miR-6531 were predicted using two miRNA databases. (**C**) The KEGG pathway enrichment of the 508 shared targets (***** *p < 0.001*).

**Figure 2 genes-13-01788-f002:**
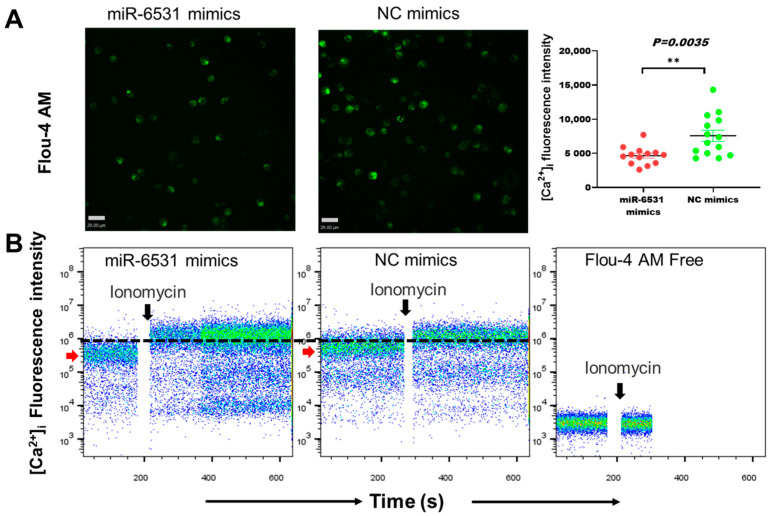
The [Ca^2+^]_i_ signaling was detected by using Flou-4 AM in bovine Leydig cells after transfecting miR-6531 mimics or NC mimics. (**A**) The [Ca^2+^]_i_ intensity test by fluorescence microscopy. (**B**) The fluorescence intensity of calcium was detected before or after ionomycin addition by flow cytometry (*** p < 0.01*).

**Figure 3 genes-13-01788-f003:**
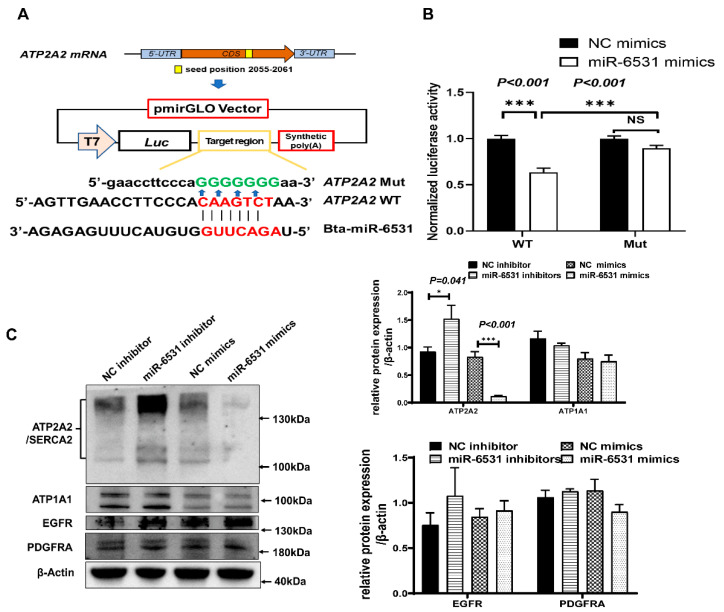
*ATP2A2* is a direct target of bta-miR-6531. (**A**) Schematic diagram illustrating the design of luciferase reporters with bta-miR-6531 binding site in wild-type *ATP2A2*-CDS region (WT) or mutant *ATP2A2*-CDS region (mut). (**B**) The relative luciferase activity was analyzed; (**C**) miR-6531 regulates *ATP2A2* protein expression without affecting the protein expression level of other predicted genes (** p < 0.05, **** *p < 0.001*).

**Figure 4 genes-13-01788-f004:**
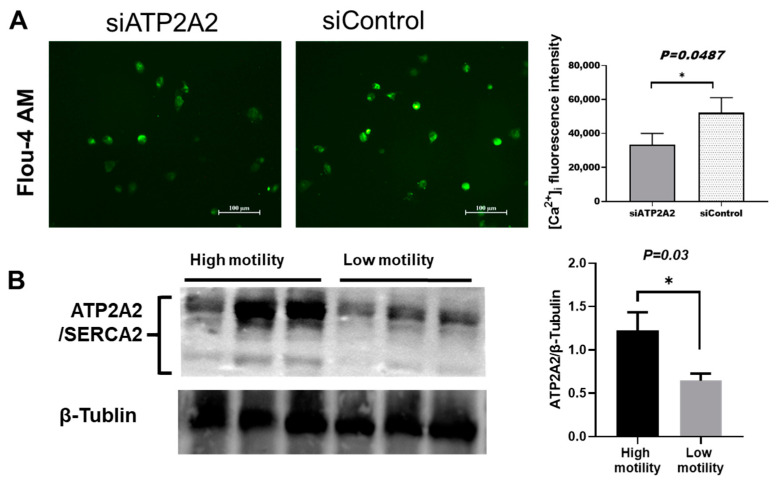
*ATP2A2* regulated calcium signaling in Leydig cells (**A**) and different expression in in high- and low- motility sperms (**B**) (** p < 0.05*).

**Figure 5 genes-13-01788-f005:**
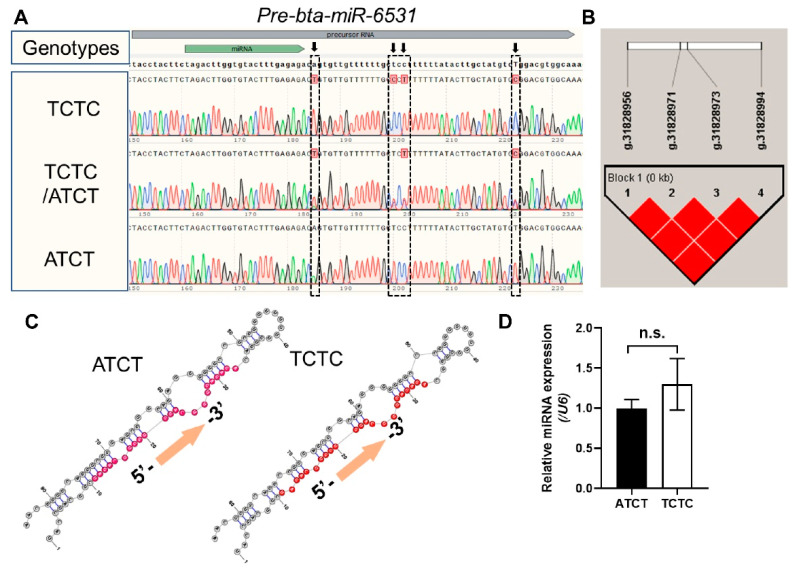
SNPs distribution within the bta-miR-6531 precursor. (**A**) The four SNPs located within the precursor sequence of bta-miR-6531. (**B**) The four SNPs showed s strong link with each other. (**C**) The four SNPs did not significantly change the structure of bta-miR-6531 precursor. (**D**) The miR-6531 expression in sperms with high and low motility.

**Table 1 genes-13-01788-t001:** Least squares mean (LSM) and stand errors (SE) for semen quality traits of different genotypes of bovine pre-bta-miR-6531.

Genotype	Allele Frequency	Ejaculate Volume (mL)	Fresh Sperm Motility (%)	Density (10^8^ mL^−1^)	Frozen Sperm Motility (%)	Deformity Rate (%)
TCTC/ATCT (*n* = 37)		6.46 ± 0.23	69.16 ± 0.55 ^ab^	12.79 ± 0.44 ^ab^	43.76 ± 1.05	16.45 ± 0.50 ^ab^
ATCT (*n* = 12)	0.54	6.05 ± 0.55	67.02 ± 1.00 ^b^	11.67 ± 0.62 ^b^	43.26 ± 1.41	17.90 ± 1.15 ^a^
TCTC (*n* = 7)	0.46	6.80 ± 0.50	70.28 ± 0.66 ^a^	14.37 ± 0.70 ^a^	44.51 ± 2.24	14.37 ± 0.57 ^b^

Note: a, b means with different small letters within the same column are significantly different (*p* < 0.05).

## Data Availability

Q.D. had full access to all the data in this study and takes responsibility for the integrity of the data and the accuracy of the data analysis. The data that support the findings of this study are available from the corresponding author upon reasonable request.

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
