# Peer review of "Bta-miR-6531 Regulates Calcium Influx in Bovine Leydig Cells and Is Associated with Sperm Motility"

_genes, 2022, doi:10.3390/genes13101788_

Round 1

Reviewer 1 Report

Bta-miR-6531 is associated with sperm quality through targeting ATP2A2 of the calcium signaling pathway in cattle sperm

The manuscript makes advances to the knowledge of the possible role of miR-6531 in the sperm quality of bulls. In the manuscript, the authors have shown that higher expression of miR-6531 is associated with lower expression of ATP2A2 and low levels of calcium in sperm cells. Even though the study has proven that ATP2A2 is a direct target of miR-6531, in sperm cells the role is shown by association. Even though the association is not a negative point of the manuscript, my concerns are regarding the over-speculation and missing of important methodological information. Regarding over-speculation, the study associated the role of miR-6531 with sperm quality. However, sperm motility was the only parameter of sperm quality that was presented and used to distinguish the samples and establish sperm quality (in addition to sperm motility, sperm quality could be characterized by other parameters such as sperm morphological defects, sperm membranes, and DNA integrity, oxidative stress, sperm activity, etc). Another important issue is related to the lack of methodological information. In that regard, sperm motility, the parameter used by the authors to distinguish the samples, was evaluated by CASA (Computer analysis of sperm motility) and, it is missing important information regarding the sperm samples used in the study (topic 2.2.), related to the number of ejaculates collected and evaluated by bulls, how many bulls were used in the study, which was the breed of the bulls, method of semen collection, age of the bulls, how were the parameters evaluated by CASA (set up, software, glass or makler chamber, the concentration of sperm during evaluation, information of the dilution of sperm - all points important to standardize the CASA analysis and to perform an appropriate sperm analysis), what were the motility threshold and CASA variables used to distinguish high motility sperm samples from low motility sperm samples, and how many samples were classified as high and as low sperm motility. Another important issue is that the intensity of calcium, the parameter associated in sperm cells with the expression of ATP2A2, in high motility in figure 4 does not seem different from low motility; it just seems that the fluorescence intensity of the image is different by the distinctive time of exposition (The authors should mention in the methodology how was performed the acquisition of the images in epifluorescence microscopy software).

Minor revision: 

Please, verify references 2 and 3 (line 43) as well as 4 and 5 (line 44). It seems that these references should be exchanged since Capra et al have shown a differential miRNA profile in sperm samples with different motility and no different fertility. Please, revise the references.

Please, verify the written in lines 87 and 91.

Reviewer 2 Report

1. L.78 – “We amplified the sequence of the pri-miR-6531 sequence”

It is advisable to prescribe primers in the manuscript

2. L. 277 – 279 “But, bulls with the TCTC genotype showed high fresh sperm motility, sperm density, and a lower deformity rate. This indicates that these SNPs are likely associated with that the  sperm have better quality.”

Only 56 bulls is a small sample for statistical analysis and conclusions

Reviewer 3 Report

Dear Authors, 

please find below few comments. 

Why did you not perform GWAS for semen quality traits?

L20 – check spelling (Holstein)

L27-28 – rephrase for increase readability  

Introduction is succinct and should be extended to attract the interest of potential readers.

The used primer sequences should be presented in the Results (2.3) or added in Supplementary.

L268 – rephrase for increase readability 

Round 2

Reviewer 1 Report

The manuscript associated the presence of miR-6531 with the expression of ATP2A2, calcium activity, and sperm quality. However, a lot of extrapolation is performed and a lot of issues should be explained in the paper before considering it for publication. The authors have answered in parts the points that I raised and serious issues continue to occur in the paper. The first one is the extrapolation of the manuscript. In the title: “Bta-miR-6531 is associated with sperm quality by targeting ATP2A2 of the calcium signaling pathway in cattle sperm”. I disagree that it is possible to conclude that miR-6531 could be related to sperm quality. Although sperm motility is an important feature that correlates with fertility, this is not the only feature used to characterize sperm quality and should not be (Please, I suggest that the authors read the following references 10.1002/j.1939-4640.1993.tb03247.x; 10.3389/fcell.2020.00791). Thus, the authors can only conclude that this miRNA is correlated with sperm motility and nothing else. Another important issue in the title and along the manuscript is related to ATP2A2 in sperm, they only showed the relation (and target as written in the title) between the miR-6531 and AT2A2 expression in Leydig cells. Thus, it is not possible to prove this relation in sperm and just to associate it. This much extrapolation needs to be strongly revised in the title and in the manuscript before considering the paper to be published in Genes and I exemplified it in the points listed below. Another important issue is regarding figures and motility. Even though the authors have standardized the exposition to acquire the images, the background doesn’t seem the same in the different images. Please, verify it. Another important issue is regarding sperm motility to separate the groups for calcium and ATP2A2. The authors need to explain better this methodology and also show the mean and SEM of motility from the groups that they compare the calcium/ATP2A2. In that regard, I don’t recommend the publication of the manuscript in the format that it is. Please, find other issues below:

1- Please, revise the sentence on line 65. It is missing information after "However...."

2- Please, revise the sentence on line 77: “Semen was packaged in 0.25ml straws after diluting with Bioxcell (IMV Biotechnology, Basse-Normandie, France), cryopreserved, and stored in liquid nitrogen for 7 days”. It is confusing and it seems that the semen was stored and then packaged.

3- Please, elucidate how the sperm deformity was considered. Was this parameter classified according to morphological defects? Which defects were considered?

4- Please, elucidate how sperm motility was considered. Was it evaluated by the same technician? Does the technician consider total motility or progressive motility? How was the dilution used for sperm motility evaluation? Was it the same for all the samples? Was sperm motility evaluated under a warm slide? All these parameters are really important to evaluate sperm motility and could alter this parameter. Since the authors based their results on this parameter, they need to provide as much information as possible.

5- Please, clarify in the methods the methodology to evaluate all sperm features (sperm motility, deformity...).

6- Please, clarify in methods and/or results how the authors separated the samples for molecular analysis (sperm motility), how many samples, and also what was the mean and SEM and P value of sperm motility of these samples. This is very important.

7- In table 1, deformity rate was evaluated in fresh or frozen samples?

8- Fig 3 (B): Please, corrected “was analyzed”.

9- Line 284: “The above results demonstrated that ATP2A2 is a negative target of bta-miR-6531 in bovine cells and may regulate calcium in sperm tails”. The evidence for that association is not directly proved in sperm cells in the paper. The authors need to be more straight forward related in their results and less extrapolation should be performed. Thus, in this part, the authors need to be more softly in relation to their conclusion. Also, in the title of the manuscript and in the conclusion. They have shown that ATP2A2 is a target of miR-6531 in Leydig cells, but not in sperm cells, thus the conclusions should be softly performed. 

10- Figure 2: provide a figure with all the samples in the same background. NC mimics and miR-6531 are not with the same background and for this, it is hard to compare both. The same observation for Figure 4A.

11- Line 320: “we have identified ATP2A2 as a functional target gene of miR-6531 by 320 targeting its CDS region” – Please, include that this was found in Leydig cells.
